# Deterministic Assessment of the Risk of Phthalate Esters in Sediments of U-Tapao Canal, Southern Thailand

**DOI:** 10.3390/toxics8040093

**Published:** 2020-10-26

**Authors:** Okpara Kingsley, Banchong Witthayawirasak

**Affiliations:** 1Faculty of Environmental Management, Prince of Songkla University, Hat Yai, Songkhla 90112, Thailand; Okparakingsley777@gmail.com; 2Research Program of Municipal Solid Waste and Hazardous Waste Management, Center of Excellence on Hazardous Substance Management (HSM), Bangkok 10330, Thailand

**Keywords:** risk quotient (RQ), standard quality guidelines (SQGs), mixture risk, uncertainty, equilibrium partition theory

## Abstract

This baseline study evaluated the ecological risk associated with the concentration of six common Phthalate esters (PAEs) in sediment samples collected from the U-Tapao canal in Southern Thailand. Deterministic approaches consisting of standard sediment quality guidelines (SQGs) and Risk quotient (RQ) were used to evaluate the potential ecological risk of individuals and a mixture of Phthalate esters (PAEs) detected in sediment samples. Of the 6 PAEs measured, only three, including di-n-butyl phthalate (DBP), di-2-ethyl hexyl phthalate (DEHP) and di-isononyl phthalate (DiNP), were identified and quantified. The total concentration of the 3 PAEs congeners found in the sediment samples ranged from 190 to 2010 ng/g dw. The results from the SQGs and RQ were not consistent with each other. The SQGs results for individual PAEs showed that DEHP and DBP found in sediment was estimated to cause moderate risk on benthic organisms, DiNP was not estimated due to lack of SQGs data. However, the RQ method indicated a low risk of DEHP and DBP on algae, crustacean and fish, whereas DiNP poses no risk on crustacean. Furthermore, based on the result obtained in this study, the consensus SQGs for mixture effects prove to be a more protective tool than the RQ concentration addition approach in predicting mixture effects. Despite inevitable uncertainties, the integration of several screening approaches of ecological risk assessment (ERA) can help get a more inclusive and credible result of the first tier of individuals and a mixture of these pollutants.

## 1. Introduction

Phthalate esters (PAEs) are essential industrial chemicals widely used in a diversity of products. PAEs are mainly used as plasticizers to improve the softness, processability, flexibility and durability of polyvinyl chloride (PVCs) products, polyvinyl acetate and polyurethane resins [1]. PAEs are not chemically linked to the polymeric matrix. Thus, they are quickly released into aquatic ecosystems through industrial and municipal discharge outlets, surface runoff from urban, agricultural and aquaculture areas, leaching from municipal solid waste sites, dumping of PAEs-containing products and atmospheric deposition [2,3,4]. On entering the aquatic environment, PAEs are distributed into various environmental media including water, suspended particles, sediments and aquatic biota, causing severe ecological risk on sensitive biota and the entire ecosystems [5]. PAEs are potential endocrine disruptors, teratogenic and carcinogenic materials that may pose adverse effects on human health, such as reproductive abnormalities [4,6]. The United States Environmental Protection Agency (USEPA) and the European Union (EU) classified some PAEs congeners as priority pollutants of the aquatic environment because they easily get attached to suspended solid particles and sediments and enter the food web [3].

Sediment ecosystems are influenced by integrating physical, chemical and biological processes, enhancing sediment capacity to maintain a functioning, active and varied aquatic population [7]. Nevertheless, contaminated sediments can be a vital source of pollution that might cause persistence degradation of the aquatic environment, even after other contaminant sources are stopped [8]. Besides, sediments are significant indicators for assessing anthropogenic pollution of chemical compounds in the aquatic environment due to their long residence time, including PAEs [9]. Currently, the contamination and ecotoxicological risk of PAEs in sediments have attracted serious attention worldwide [10,11,12,13]. Several studies have reported the elevated PAEs pollution that poses severe adverse effects on the aquatic ecosystem [12,13,14]. Moreover, Water Framework Directives (WFD) categorically support the inclusion of sediments in the risk assessment of priority pollutants in the aquatic ecosystem [15]. However, there are limited studies on the contamination and potential ecological risk of PAEs in sediments in developing countries, including Thailand [14]. Studies on the contamination and potential ecological risk of PAEs in Thailand sediments are scarce, except in the Chao Phraya River and the Eastern coast of Thailand [16,17].

The U-Tapao canal, the mainstream freshwater resource of a significant watershed, is located in a tropical region in Songkla Province, Southern Thailand. The canal originates from Bantad mountain and flows through Hatyai city before emptying into Songkhla Lake. Like other water bodies, the canal is frequently exposed to both point and non-point environmental pollution sources due to rapid economic development, urbanization and changes in land-use patterns in the area surrounding the water body. The primary water quality degradation sources of the freshwater are wastewater effluents discharge from rubber, plastics, parawood and seafood processing industries at the rate of 41,000 m^3^/day, which have negatively impacted the ecological integrity of the canal [18,19]. Besides, we observed levels of PAEs that pose ecological risks in surface water of U-Tapao [20]. Nevertheless, the levels of PAEs in sediments is still unknown. To protect the aquatic ecosystem of the U-Tapao canal, it is imperative to determine the level of PAEs in sediment and their subsequent potential ecological risk on aquatic organisms.

Ecological risk assessment (ERA) aims to evaluate the likelihood and the extent of adverse effects due to exposure to pollutants or other stressors [8]. ERA conventional methods are deterministic approaches, such as sediment quality guidelines (SQGs) and Risk Quotient methods, suitable for the preliminary stage of risk assessment [7,21,22]. The SQGs and RQ approaches can be used by risk analysts and other decision-makers to assess the preliminary ERA of pollutants, whether the value exceeds any predetermined threshold levels of concern. To get a more credible result, some researchers and scientific institutions have recommended using a comparative approach for ecological risk assessment of pollutants [23]. Comparative analyses can ensure consistency in risk management decisions and focus more on the significant ecologically-based risk management decisions on contaminants that pose the highest risk to benthic biotas, algae, crustacean, fish and the entire ecosystem [23]. Systematic and comparative study of the predictive abilities of SQGs and RQ approaches for PAEs in sediments is lacking.

Therefore, the primary objective of this study was to evaluate the toxicity induced in aquatic organisms by individuals and a mixture of PAEs congeners, namely DBP, DEHP and DiNP, in sediment samples collected from the U-Tapao canal by using the deterministic approaches, including SQGs and RQ. The importance of these pollutant contaminations will be measured in terms of their possible effects on the benthic community and sensitive trophic biota, including algae, crustacean and fish, determining which areas of the canal will be of particular concern, thus needing further investigation. Furthermore, the comparative ERA of the individuals and the mixture of PAEs in the freshwater ecosystem located in a tropical region can contribute to the definition of freshwater sediment quality guidelines in the Water Framework Directive (WFD).

## 2. Materials and Methods

### 2.1. Study Site and Design

The study area and sampling sites are indicated in Figure 1. To assess the extent of PAEs contamination and potential risk in sediment, a cross-sectional study was conducted in the U-Tapao canal. The freshwater ecosystem is 68 km long and approximately 3 m to 8 m deep. It originates in the southern mountainous areas, flows northward through the watershed center and drains into the Songkhla Lake. The canal’s flow rate ranges between <6 and 90 m^3^ in dry and rainy seasons. Two monsoons strongly influence the river’s tropical monsoon climate; the northeast and southwest monsoon, with average rainfall estimated to range from 1600 mm to 2400 mm annually. The temperature within and around the riverine ecosystem varies between 24 °C and 32 °C all through the year.

In this present study, seventeen sampling sites of sediments were selected along the canal, from the upstream to downstream. Sampling points were identified by using a global positioning system (GPS). On-site sampling of all the surface sediment was performed from August 2018 to March 2019 at 17 locations in the U-Tapao canal. These locations were classified into two different groups viz: urban and rural areas. Sampling sites in urban areas include ST1, ST2, ST 3, ST4, ST6, ST7, ST9, ST10, ST12 and ST13. Sites located in the vicinity of the rural area were ST5, ST8, ST11, ST14, ST15, ST16 and ST17. Sediment samples were collected by using a grab sampler and transferred onto pretreated wide-mouthed brown glass bottles. The bottles were immediately submerged in crushed ice and transported to the laboratory. The samples were freeze-dried for 72 h, ground to pass through a 0.5 mm sieve, thoroughly homogenized, dried, and kept in a deep freezer at −22 °C. All sediment samples were analyzed within three days.

### 2.2. Chemicals and Material

Solvents used for this work include high performance liquid chromatography (HPLC) grades of hexane, methanol, acetone, ultrapure water and dichloromethane (Waters, Milford, MA, USA). Phthalate standards included di-n-butyl phthalate (DBP), benzyl butyl phthalate (BBP), di-2-ethyl hexyl phthalate (DEHP), di-n-octyl Phthalate (DnOP), di-isononyl phthalate (DiNP), diisodecyl phthalate (DIDP) (AccuStandard Inc, (New Haven, CT, USA). Internal standard solutions, including phenanthrene-d10 and chrysene-d12 and standard surrogate solutions, 2-fluorobiphenyl and 4-terphenyl-d14, were obtained SUPELCO Inc. (Bellefonte, PA, USA).

### 2.3. Pretreatment in Sediments

The freeze-dried sediment samples collected from the U-Tapao canal were pretreated based on a method developed [24] with slight modification. Every five grams of sediment samples were crushed and homogenized using a mortar and pestle and filtered via a stainless-steel sieve (60-mesh) and placed in brown glass bottles at −20 °C pending extraction. Weighed sediment samples (5.0 g) were placed into clean glass centrifuge tubes, mixed with 10 mL acetone/hexane (1:1 *v*/*v*) and 0.2 mL of 10 mg/L mixture of standard surrogate solutions (2-Fluorobiphenyl and 4-Terphenyl-d14). A procedural blank not containing the sediments were also prepared by using a similar procedure; 1:1 (*v*/*v*) acetone/n-hexane was used to prepare a check standard mixture. The spiked sample was made by mixing a standard mixture with a sediment sample. All samples were vortexed for 1 min and ultrasonicated for 20 min. The samples were further centrifuged at 3000 revolutions per minute (rpm) for 10 min. After centrifugation, the organic layer containing the extracted PAEs was siphoned out and placed in tubes using a Pasteur pipette and the process was repeated twice with 10 mL of a 1:1 (*v*/*v*) acetone/*n*-hexane. The extracts were pooled together. Desulphurization was achieved by adding activated copper to the extract. The extract was further dried over anhydrous sodium sulfate, concentrated to 0.8 mL using a gentle stream of nitrogen, added to 0.2 mL of 5 mg/L internal standard (Acenaphthene-d10, Phenanthrene-d10 and Chrysene-d12) mixture solutions and analyzed using gas chromatography (GC) with the mass selective detector (MSD).

### 2.4. Instrumental Analysis by GC-MS

All samples were evaluated using a gas chromatograph/mass spectrometer (GC–MS), Agilent model 6890N GC–5973 MSD (Agilent Technologies, Santa Clara, CA, USA), functional electron influence as well as a selective ion monitoring mode with an HP-5 MS (30 m × 0.25 mm × 0.25 mm). Chromatographic separation was performed by using the fused-silica capillary column. Pure helium gas (99.9999%) was used as the carrier gas and was maintained at a constant flow rate of 1 mL/min. The temperature program column oven was set to 30 °C for 1 min, raised to 280 °C at 15 °C maintained for 1 min, then increased up to 310 °C and held for 4 min. Each extract volume of 2.0 µL was injected into the GC–MS system in non-pulsed and splitless mode with an injector temperature of 290 °C. MS was operated in full-scan mode from *m/z* 35–500 for qualitative analysis. Acquisition for quantitative analysis was carried out in the single-ion monitoring mode along with target ions, as indicated in Table 1. The levels of PAEs in the sediments were normalized to a dry weight (dw) basis. The sample chromatograms are shown in Appendix A.

### 2.5. Quality Control and Quality Assurance

All sampling equipment comprises of glass or stainless steel. Amber glass bottles were thoroughly washed with laboratory-grade detergent, cleaned twice with an HPLC grade of acetone, hexane and dichloromethane and then heated in a muffler oven at 400 °C for at least ten hours. After baking, the bottles were pre-rinsed three times with acetone, hexane and dichloromethane, then covered with clean aluminum foil. Before their usage, aluminum foils were also rinsed in acetone and hexane and heated in a hot oven at 350 °C for ten hours. Stainless steel sampling utensils such as spoons, flat trays and buckets were washed and wrapped with aluminum foil before sampling. The sediment grab sampler and glass water samplers were cleaned with lab-grade detergent and then passed three times with an HPLC grade of acetone, n-hexane and dichloromethane, respectively. Mortars and pestles were cleaned using the same procedure as that for glassware but were baked at the 150 °C for ten hours. Besides, to ensure that the results obtained in this work are reliable, various techniques were employed, including the development of calibration curve, usage of procedural blank, the establishment of the lower limit of detection (LOD) and Limit of quantification, assessment of the precision; and the calculation of recovery percentage. The instrument was calibrated daily by preparing a calibration curve at five different concentrations. All procedural blanks’ values were less than the detection limits. For the various PAEs congeners, limits of detection (LOD) and limit of quantification (LOQ) for individual PAEs were assessed based on a signal-to-noise ratio of 3 and 10 times. Recovery efficiencies for the surrogate standards are between 81 (2-Fluorobiphenyl), 93 (4-Terphenyl-d14) and the average recovery efficiencies for the spiked samples are between 81 and 105%. The percent recovery, precision (% RSD), LOD and LOQ of the six targeted PAEs are shown in Table 1.

### 2.6. Ecological Risk Assessment of PAEs in Sediments

#### 2.6.1. Sediment Quality Guidelines (SQGs)

Pollutants in sediments can adversely affect benthic organisms via direct toxicity [25]. There are currently no generally recognized sediment quality criteria; thus, numerical sediment quality guidelines (SQGs) proposed by several international agencies and researchers have been used to assess the ecological risk posed by specific pollutants [26]. PAEs that are bio-available may pose a severe environmental risk on benthic biota associated with the U-Tapao canal and cause human health risk [14]. SQGs have been useful in assessing PAEs related to ecological risk via sediments in aquatic environments [12,24,27].

In this present study, empirical SQGs were used in evaluating the ecological risk of PAEs to benthic organisms, including the effect approaches such as threshold effect level (TEL), the probable effect level (PEL), new sediment quality guideline quotient (NSQGQs) [28,29]. Also, the consensus approach, such as the threshold effect concentration (TEC), medium effect concentration (MEC), the probable effect concentration (PEC) and the probable effect quotient (PEC-Q). As well as the equilibrium partition approach, including the maximum permissible concentrations (MPCs), the ecotoxicological serious risk concentrations (SRC_eco_) and the environmental risk limits (ERLs). Since the values of MPCs, SRC_eco_ and ERLs have been set on a 10% OM-normalized basis, the original DEHP concentrations and DBP levels were divided by the OM content (%) and multiplied by 10 to compare it with the SQG (MPCs, SRCeco and ERLs) [30,31]. The NSQGQs, PEC-Q, and mean PEC-Q_PAEs_ were evaluated by using Equations (1)–(3) below:(1)NSQGQi=(Ci/TEL+(Ci/PEL))22.

*NSQGQ* estimates both *PELs* and *TELs*. *C_i_* represents the concentration of pollutants in sediments, *TEL* is the threshold effect level of contaminants and *PEL* is the probable effect level.
(2)$$PEC-Q_{individual}=\frac{C}{PEC}.$$

*C* is the concentration of the pollutant in sediment in dry weight, *PEC* is SQG concentration of chemicals in dry weight. The mixture effects of measured PAEs by using consensus SQGs were performed by using Equation (3) below:(3)$$Mean\;PEC-Q_{PAEs}=\frac{\sum^{\;}individual\;PAEs\;PEC-Q_{i}}{n}.$$
n is the number of compounds for which *PEC-Qi* was estimated.

#### 2.6.2. Risk Quotient Method

The Ecological risk assessments of PAEs were also carried out by using the risk quotient method (RQ) following the European Commission’s Technical Guidance Document [21] and previous studies [22,32]. RQ for detected PAEs congeners was estimated by dividing the measured environmental concentration (MEC) with the predicted no-effect concentration (PENC), as indicated in Equation (4). In this study, the mean and highest MEC values were used for calculating general RQmean and worst-case scenarios RQmax, respectively. The risk quotient of the mixture (RQ_mix_) of PAEs was calculated based on the summation of individual RQ values of PAEs congeners detected in sediments, as indicated in Equation (5).
(4)$$RQ=\frac{\mathrm{MEC}}{\mathrm{PNEC}}$$
(5)$$RQ_{mix}=\sum_{i=n}^{n}RQ_{i}.$$

The criteria used to assess the ecological risk assessment were RQ > 1 (the log Kow of PAE congener between 3 and 5) showed high risk, whereas RQ > 10 (log Kow > 5) suggested high risk [22]. The US EPA ECOTOX database provided the chronic or acute toxicity data of PAEs to the algae, crustaceans and fish for this study (http://cfpub.epa.gov/ecotox/). The Equation S1, based on the mechanistic or EqP approach, was used for calculating PNEC values in the sediment and listed in Appendix A.

### 2.7. Analysis of Sediment Organic Matter

5 g of sediment sample was used to determine the organic matter levels in each subsample of sediments collected from the U-Tapao canal. The samples were oven-dried at 105 °C for 8 h to obtain a constant weight. After drying, the samples were baked in the furnace at 550 °C for 5 h and the OM level was obtained by measuring the weight loss [33].

### 2.8. Statistical Analysis

Statistical analysis was performed with SPSS version 20.0 (IBM SPSS InC., Chicago, IL, USA).

## 3. Results and Discussion

### 3.1. The Environmental Concentration of PAEs in Sediments

The statistical summary of PAEs measured in sediment samples collected from the U-Tapao canal is shown in Table 2. Of the six targeted PAEs, including DBP, BBP, DEHP, DnOP, DiNP and DIDP), only three congeners were detected, such as DEHP, DiNP and DBP. The total PAEs concentrations in the samples ranged from 190 to 2010 ng/g dw, with a mean value of 899.71 ng/g dw. The most abundant PAEs congener was DEHP ranges from 190 to 890 ng/g dw, followed by DiNP ranging from non-detectable (ND) to 840 ng/g dw, then DBP varies from ND to 280 ng/g dw. The average environmental concentration of these individual PAEs was 484.24, 332.65 and 88.82 ng/g dw for DEHP, DiNP and DBP, respectively. In this study, the frequency of detection of individual PAEs followed DEHP > DiNP > DBP. This finding is in agreement with the observation of a previous study [22]. The preponderance of DEHP in sediment is attributed to high production and consumption volume, strong sorption and low degradation rate [2,6]. However, to reduce human health risk and environmental risk, DEHP usage was restricted by regulation and replaced by DiNP. Thus, it is no wonder that DiNP was also found in high concentrations in investigated sediments. This result is consistent with previous studies [34,35]. It is, therefore, recommended to include DiNP when screening aquatic sediments for PAEs.

### 3.2. Correlation between the OM and pH of Sediments and PAEs Concentration

In this research, the Spearman correlation was used to assess the relationship between pH, OM, DBP, DEHP, DiNP and total PAEs concentrations in sediment samples of the U-Tapao canal. As shown in Table 3, the spearman correlation matrix showed a significant correlation (*p* < 0.05 and 0.484) between the levels of DEHP and OM, suggesting that the spatial distribution of DEHP in this study was not only controlled by sediment OM but also depended on other factors, such as transport, mixing and sedimentation mechanisms and source compositions, which is consistent with other studies [10,11]. However, there was no significant relationship between other PAEs (DBP, DiNP and ∑PAEs) and OM. This observation indicates that the PAEs in water and sediment phases are not at equilibrium [36]. Besides, the variances of PAEs content in sediments are attributed to different pollution sources, such as various factories, residences, commercial areas and discharge outlets. The pH in sediment samples ranged from 5.73 to 8.21, with an average pH of 6.88. Moreover, ΣPAEs and individual PAEs concentrations were not correlated with pH, respectively, suggesting that pH was not a significant parameter controlling sediment-associated PAEs in this study. This is consistent with previous studies [10,11,37]. Furthermore, DBP, DEHP, DiNP showed significant positive correlations (*p* < 0.01) to ΣPAEs concentration in sediments of the U-Tapao canal. Spearman’s correlation coefficient was 0.675, 0.759 and 0.854 between ΣPAEs and DBP, ΣPAEs and DEHP and ΣPAEs and DiNP, respectively. This suggests that DBP, DEHP and DiNP are representative of ΣPAEs levels in sediments of the U-Tapao canal. However, the most substantial relationship (*p* < 0.01 and 0.854) was observed between ∑PAEs and DiNP, indicating that this congener can be a more suitable marker pollutant for detecting ∑PAEs in sediment samples. Moreover, these imply that these contaminants might have the same sources. Owing to the physical and chemical properties of PAEs, namely low-water solubility, high organic carbon–water partition coefficients (Koc) and hydrophobicity, PAEs tend to adsorb onto suspended particles and eventually accumulate in the sediment as particles settle out of solution [38].

### 3.3. Comparison with Other Studies in the World

As shown in Table 4, a comparison of available data on PAEs in sediments in Thailand revealed that the concentrations of PAEs, particularly DEHP in the U-Tapao canal, were below the levels in Chao Phraya River and the Eastern coast of Thailand [16,17]. However, overall, the PAEs concentration measured in the canal is relatively moderate compared to values in other locations in the world. The sedimentary DBP concentration in this work was higher than those in Kaoshiung Harbor, Taiwan and Jiulong River estuary, China [35,39], similar to the Jiulong River [22], while significantly lower than those in the Dianbao river Taiwan, False creek Vancouver, Canada and Chiangjiang river estuary, China [40,41,42]. DEHP concentration in this study was significantly lower than those observed in False creek Vancouver, Canada, Dianbao River and Chiangjiang River estuary, China [40,41,42]; comparable to the levels in Jiulong River, China [22] while higher than those in the Jiulong River estuary [39]. Very little information on DiNP in freshwater sediments are available in the published literature. DiNP concentrations in this work were higher than those in the Jiulong River and Jiulong River estuary, China and Cianjhen River in Taiwan [22,39] and significantly lower than levels measured in False creek Vancouver, Canada, Kaoshiung harbor and Dianbao river, Taiwan [35,40,41].

### 3.4. The Potential Ecological Risk of PAEs on Benthic Organisms

The effect characterization of PAEs in sediment samples was based on several SQGs values reported by international and national agencies and in the literature.

Based on the DEHP empirical SQGs value documented by Reference [25], DEHP TEL and PEL values were obtained as 182 ng/g and 2647 ng/g, respectively. The DEHP concentration below the TEL value indicates that the level of pollutants may not pose an adverse biological effect on benthic organisms, while those between TEL and PEL can pose moderate adverse effects and the DEHP level higher than PEL can cause severe biological effects on benthic biotas. As shown in Table 5, the results revealed that 100% of the sampling sites had DEHP values >TEL and <PEL, with DEHP concentration in sediments measuring approximately 1–4 folds higher than TEL value, suggesting chances of generating estimated moderate adverse risk on benthic organisms. A new sediment quality guideline quotient (NSQGQ), as developed by [43], was used to further assess the potential adverse biological effects of DEHP in sediments. As shown in Table 5, all the NSQGQs values obtained for all the sampling sites ranged from 0.51 to 1.16, which indicates that DEHP is posing moderate adverse biological effects on benthic organisms. DBP and DiNP were not evaluated by this method due to a lack of standards values.

Several derived empirical SQGs values reported for pollutants exhibit a high variability; thus, some authors developed the consensus-based approach to harmonize the existing values [26]. This approach was also applied to evaluate DBP and DEHP’s potential adverse biological effects on benthic organisms in this study. As indicated in Table 5, data obtained in this work shows that the DBP concentrations were well below the TEC, MEC and PEC values, indicating an unlikely presence of adverse biological effects for the benthic organisms. However, for DEHP, 29% of the sampling sites, including sites 1, 2, 4, 13 and 16, were higher than the TEC value, suggesting the incidence of sediment toxicity in these sampling sites. Besides, area 13 was more elevated than MEC values, indicating more significant sediment toxicity in this site, whereas PEC value exceeds the concentration of DEHP in all the sampling sites, showing the absence of severe sediment toxicity in 100% of the sampling sites. Besides, the individual PEC-Q values of DBP in all the sampling sites ranged from 0.002 to 0.017, indicating 0% of sediment toxicity on benthos, whereas the individual PEC-Q values of DEHP ranged from 0.17 to 0.81, suggesting sediment toxicity ranging from 0% to 40% on benthic organisms. Since the incidence of toxicity has been described as dependent on the magnitude of the PEC quotient values [26,34], those areas with higher PEC-Q concentration are most probably associated with adverse effects on the ecosystem. It is worthy to note that almost all the sites with higher TEC, MEC and PEC-Q are located in the urban areas, consistent with previous studies [26]. Because the TEC, MEC and PEC data are not available for DiNP, the potential sediment toxicity of DiNP was not assessed by this approach.

Alternate methods such as the equilibrium partitioning theory by Reference [28], have been widely employed by many researchers to evaluate the ecological risk posed by PAEs, at 10% organic matter normalization [12,13,32,35]. As indicated in Table 6, application of this method based on data obtained in this study shows that DBP concentrations were >MPC in 41% of the sampling sites and well below SRCeco value in all the sampling sites, suggesting that the level of DBP in sediments may pose a moderate ecological risk on the benthic organism in the polluted sampling sites. For DEHP, 100% of the sampling sites were >MPC and <SRCeco; similarly, for ∑PAEs, 100% of the sampling stations were >MPC and <SRCeco, suggesting that DEHP and ∑PAEs are considered to be posing a moderate ecological risk in benthic biotas. Because the MPC and SRCeco data are not available for DiNP, the potential risk of DiNP was not assessed. However, previous studies indicated that if the pollutant level exceeds the Environmental risk level (ERL) value, it may generate several adverse effects on environmental and human health [10,11,12,13]. ERL concentrations of DBP and DEHP capable of causing an adverse ecological impact on an ecosystem have been proposed to be 700 ng/g and 1000 ng/g, respectively, in sediments containing 10% organic matter [27]. AS shown in Table 6, the DBP concentrations in 60% of the sampling sites were >ERL, whereas 40% of the sites were <ERL, indicating that DBP maybe posing adverse effects in 60% of the sampling sites. For DEHP, 100% of the sampling sites were >ERL, suggesting that DEHP poses an adverse ecological risk to the benthic organisms in all the sampling sites. Our result is in agreement with the ERLs reported from the eastern coast of Thailand [17]. The ERL of DiNP was not evaluated due to a lack of data; thus, the environmental risk of DiNP is unknown.

Overall, the SQGs results showed moderate sediment toxicity to benthic organisms in the freshwater canal ecosystem. Sediments are very useful to demersal bivalves as they play a dual role as habitats and a source of food through direct ingestion and indirect exposure [44]. Contaminated sediments have been indicated to be responsible for accumulating pollutants in aquatic biotas’ tissues, which may eventually affect both humans and other wildlife that consumes aquatic organisms [25]. These moderate PAEs concentrations in the canal sediments might expectedly cause changes in the microbial communities and enzyme activities and affect water quality [45]. However, the occurrence of PAEs in sediments used as organic fertilizers may result in PAE accumulation in crops or vegetables, impacting human health via the food chain [37]. Thus, controlling PAE concentrations in the canal sediments used in aquaculture or agricultural purposes is extremely important in reducing these organic pollutants’ environmental health risks. Therefore, we recommend that the responsible agencies commence effective mitigation measures to control PAE emissions into the U-Tapao canal to prevent further PAEs pollution and protect the quality of the canal environmental health status.

### 3.5. The Potential Ecological Risks of Individual PAEs to Algae, Crustacean and Fish

Furthermore, to understand the environmental risk of PAEs to the major trophic organisms, including primary producers, primary and secondary consumers, inhabiting U-Tapao canal, surrogate species of algae, crustaceans and fish were selected and treated separately in the calculation of the risk quotient (RQ) for detected PAEs [22]. The toxicity data of individual PAEs for each trophic level of sensitive aquatic biota, coupled with PNEC_sediment_ of individual PAEs for these three groups, are presented in Appendix A. RQ of individual PAEs was calculated using mean and maximum values of the measured environmental concentration of PAEs in Table 2 divided by the predicted no-effect concentration in sediments (PNEC_sediment_) in Appendix A. As shown in Table 7, the RQ values of individual PAEs in the freshwater sediment samples followed the order: DEHP > DBP > DiNP. The RQ mean and maximum values of DEHP (log Kow = 7.3), DBP (log Kow = 5.6) and DiNP (log Kow = 8.8) were <10, suggesting low risk to the sensitive aquatic biota, including algae, crustacean and fish living in the canal ecosystem. However, this result is consistent with the findings in previous works that reported a low risk of DEHP, DBP and DiNP in sediments to sensitive aquatic biota in freshwater environments [5,22].

### 3.6. Mixture of Ecological Risks of PAEs by SQGs and RQ Methods

In an aquatic ecosystem containing mixtures of PAEs at specific concentrations, the concentrations of some individual PAEs are so low that their specific adverse biological effects may not be apparent if applied individually [46]. However, their combined adverse biological effects on benthic organisms and other sensitive aquatic biotas, including algae, crustacean and fish, will remain [46]. Using the relationships derived from existing databases, the mean PEC-Q_PAEs_ value can be used to predict the toxicity of a mixture of contaminants in sediment. Mean PEC-Q_PAEs_ enables risk assessors to estimate the probability that contaminated sediments will be toxic to sediment-dwelling organisms [26,34]. Based on the consensus approach of SQGs, the mean PEC-QPAEs was used to assess the combined effects of DEHP and DBP measured in sediment samples; the results obtained were 0.5 and 0.8 for average and maximum concentration of DBP and DEHP mixture respectively, suggesting that the average mixture concentration of DBP and DEHP could cause 40% incidence of sediment toxicity. Whereas, the maximum mixture concentration can pose 54% toxicity via sediment. Therefore, developing the empirical SQGs values for several PAEs congeners is imperative and will facilitate evaluating the mixture risk of PAEs, using the SQGs approach.

As shown in Table 7, the RQmix value for RQmean and RQmax was well below 10. The highest RQmix values were 1.9 × 10^−1^ and 3.9 × 10^−1^ for RQmean and RQmax on algae and the lowest RQmix values were 7.0 × 10^−2^ and 1.8 × 10^−1^ on crustacean. However, the RQmix values were approximately 5 and 2 times higher than the individual risk of DBP and DEHP, respectively, for Algae. For crustaceans, the cumulative RQ value was two times higher than the individual risk of DBP and DEHP and 68 times higher than DiNP. For fish, the total RQ value was six and 1-fold more elevated than the individual risk of DEHP and DBP, respectively.

Nevertheless, overall results based on the RQ approach indicate that the mixture effects of DEHP and DBP in the sediments would cause a low or no adverse impact on algae and fish and a crustacean. However, the sediment can be an extended/archived source of anthropogenic compounds in the aquatic environment [47]. Since PAEs are one of the pseudo-persistent pollutants, its prolonged exposure may cause plausible health effects in fish and other non-target aquatic species [48]. Moreover, ecological combined effects of PAEs can generate serious concern [49]. A study demonstrated that a combination of BBP, DBP, DEHP, DiNP, DIDP and DnOP caused high adverse effects and induced severe developmental toxicity in live embryos of aquatic Zebrafish [50]. Hence, further research on the potential combined ecotoxicological effects of PAEs mixtures in the canal ecosystem is required. The lack of ecological risk assessments of pollutants may adversely affect environmental monitoring and management efforts. Currently, environmental regulations and policy on PAEs in the aquatic environment in most developing countries are lacking. This, in turn, impacts the ecological quality and public health. Besides, exposure and risk assessments are vital to prioritize risks in the guidelines and standards for significant decision making for risk mitigation [51]. SQGs and RQ approaches are the most commonly used methods in the screening stage of ERA. The main advantage of these approaches includes simplicity, transparency and low data requirements. However, this approach cannot account for the spatial-temporal variability of exposure levels and the probability and magnitude of ecological effects [51]. Thus, the probabilistic method should be applied to assess PAEs in the U-Tapao canal further to solve this problem.

### 3.7. General Comparison among SQGs Approach and Assessment of the Toxicity Induced by PAEs

Because the SQGs values used in this study were not derived from this region, the degree of variability in the use of SQGs may be attributed to regional differences in the geochemistry of sediments and the relative bioavailability of sediment-associated toxicants [52], should be tolerable. Moreover, the application of the SQGs to regions other than those for which the values were developed should require further on-site toxicological data to clarify the degree to which adverse effects towards sediment-dwelling organisms will be produced [53]. The dearth of this data led us to consider that the results obtained with SQGs used in this work are not intended for management purposes but only for setting of benchmark levels for identifying the areas of the U-Tapao canal that may be of particular concern about the present and future sediment contamination by PAEs.

The comparison with the different SQGs as shown in Table 5 and Table 6, the upper limit of the TEL/PEL approach, which is PEL is entirely comparable to the upper limit of TEC/PEC, called PEC and MPC/SRCeco, namely SRCeco, since the level of PAEs measured at 100% of the sediment samples were well below all the values of these upper limits. The result obtained by using the RQ approach was comparable with those of the upper limits of the SQGs. Thus, severe sediment toxicity can be considered unlikely for the sediments from the U-Tapao canal. However, the lower limits such as TEL, TEC and MPC, were also a perfect comparison, especially between TEL and MPC, because 100% of the sediment samples were well above the TEL and MPC values. DEHP is the dangerous primary pollutant to the sediment-dwelling biota, exceeding the TEL and MPC in 100% and TEC in 29%, whereas DBP concentrations were well below the MPC values TEC. The high SQGs values (TEL, MPC and TEC) may be attributed to the discharges of untreated and semi treated effluent from rubber and plastic industries into the canal [18,19,54].

As shown in Appendix A, individual PEC-Q and NSQGQs for DEHP exhibited a significant linear correlation (R2 = 0.990, *p* < 0.01), indicating that NSQGQs could be an acceptable method to evaluate the sediment toxicity of DEHP. Compared with NSQGQs, the risks of DEHP for some sediment samples were underestimated using the technique of individual PEC-Q. For example, 29% of the canal sediment samples were identified as no effects based on the individual mean PEC-Q for DEHP, while these samples were suffering from moderate biological effects according to NSQGQs. Thus, NSQGQs established in this study could better evaluate the ecotoxicological risks of DEHP.

In contrast, the consensus TEC approach seems less conservative and protective for the U-Tapao canal because only five samples (29%) showed DEHP levels higher than TEC invariance to 100% obtained for MPC, ERLs and TEL, respectively (Table 6). The differences noticed for these SQGs with the TEC are mainly due to the 10% organic matter normalization used in calculating MPC and ERLs. Generally, concerning false-negative and false-positive classifications, when chemical concentrations are lower than TEC values, the probability of samples correctly classified as non-toxic is higher than 75% [25,26]. In contrast, those sediments whose concentrations were between TEC and PEC values represent sediments in which toxicity is possible but it is uncertain to what extent adverse effects could appear [25,55].

The mean PEC-Q_PAEs_ was used to assess the toxicity induced by the mixture of the two PAEs congeners (DBP and DEHP) with TEC and PEC values indicating that sediments from the nine sites (ST1, ST2, ST3, ST4, ST9, ST10, ST13 and ST16) might be inducing sediment toxicity to aquatic organisms in the canal. Besides, ST13, which is located in the Hat Yai city, exhibited the highest potential toxicity to benthic organisms and relatively high sediment toxicity occurring in sampling sites ST4, ST2, ST1 and ST3, located close to high anthropogenic areas. On the contrary, the sites ST14 and ST15, located near aquaculture ponds, showed a negligible toxic effect of the target chemicals due to the impact of surface water dilution and ST16 and ST17, found in the less anthropogenic area of the U-Tapao canal.

Figure 2 shows the spatial distribution of the sediment toxicity trend of the mixture risk of DBP and DEHP detected in 17 sampling sites in the U-Tapao canal, with 53% of the areas having a mean PEC-Q_PAEs_ higher than 0.25, which indicates a 20% probability of toxicity in these sites. This may be attributed to the discharge of domestic effluent, industrial wastes, urban runoff, the outflow from agricultural areas, atmospheric deposition and plastic products’ indiscriminate disposal. Thus, an ecological change is evolving in this canal due to frequent discharges of untreated domestic and industrial effluents.

In the ecosystem with intense agricultural and aquaculture activities and many industrial and municipal sources of chemical inputs, such as the U-Tapao canal, it is difficult to pinpoint a clear source-occurrence relationship. Moreover, the sediment resuspension from tidal movements in the canal may promote transport, further masking source-concentration matches, as was indicated by the contamination revealed at site ST 12 and 14, which was lower than that observed at the close sampling sites ST11 and 15, respectively. Again, the canal’s hydrological characteristics, such as macro-tidal effects and seasonal fluctuations in flow-rates due to profound impacts of the Southwest Monsoon (annual rainfall 1600 mm), can influence the distribution of these compounds in sediments [18,19,54].

The sampling site (ST16) was the only site located in the rural area of the canal with a mean PEC—Q_PAEs_ higher than 0.25, indicating 20% of sediment toxicity, while the close sampling sites ST 17 showed a low mean PEC-Q value of 0.13, which corroborated the insignificant effect on the sediment-dwelling organisms. The threshold level was exceeded at ST 16 due to the relatively high DEHP levels, which also exceed TEL, ERLs and MPC values in 100% and 29% for TEC.

Notwithstanding that ST 12 showed MPC, ERLs and TEL threshold value violations, the calculated mean PEC-Q_PAEs_ was estimated at 0.10, indicating the negligible toxic effects of the PAEs mixture in sediment. This situation represents a typical example in which SQGs contrasts with a simple comparison between chemical data.

### 3.8. Uncertainty Analysis of Deterministic Risk Assessment by SQGs and RQ Approaches

ERA’s limitations are expected irrespective of the risk assessment approach used, whether deterministic (qualitative) or probabilistic (quantitative). The uncertainties in ERA are associated with an irregularity in ecosystem stressors, exposure data, species effect data, type of model and knowledge gap [51,56]. For instance, the contamination and distribution of PAEs data in the U-Tapao canal are limited. Except for our six-targeted PAEs concentration data measured in the canal ecosystem, there are no additional PAEs concentration data available for this aquatic ecosystem. Inadequate data of PAEs are an essential source of weakness in exposure assessment. Further work should be done to get more PAEs data in various spatial and temporal scales.

Empirical SQGs are obtainable for a few numbers of PAEs congeners. For instance, the only PAEs congener with TEL and PEL value is DEHP, ERLs values are available for DEHP and DBP, whereas TEC and PEC values are available for DBP, DMP, DEHP and DnOP. Overall, empirical SQGs are available for 4 PAEs congeners, representing a minute fraction of the total PAEs extensively used globally. Underestimation of the ecological risk of PAEs in sediment may be attributed to the unavailability of SQGs data for several other PAEs. Empirical SQGs predict thresholds at which a toxicity response of benthic animals is probable.

Nevertheless, SQGs, in general, do not provide a perspective on the risk and possible adverse outcomes associated with toxic concentration levels, which necessitates a separate ERA. Concentration ranges between upper and lower empirical SQG values are generally significant. The higher the variation between these values, the more the extrapolative value and sensitivity of the SQGs are compromised. For instance, the PEL value for DEHP is approximately 15 times higher than the TEL value. Besides, TELs are derived from the lower 15th percentile of effects data and the 50th percentile of no effects data, whereas PELs are derived from the geometric mean of the 50th percentile of the 85th percentile of the no effects data.

Equilibrium partitioning (EqP) SQG, including ERL, MPC and SRCeco, predicts the concentrations of PAEs based on equilibrium conditions. Besides, functional relationships do not model contaminants’ release trapped in pores of sediments where they can be sorbed on interstitial walls [7]. The EqP approach assumes that sediment toxicity’s critical factor is contaminants’ concentration in sediment pore water and ignores exposure through sediment and food ingestion. Porewater pathways may be unlikely for hydrophobic pollutants such as PAEs, which are low in the dissolved phase

The RQ approach cannot account for the spatial-temporal variability of exposure levels and the probability and magnitude of ecological effects. To address these limitations of this method, a probabilistic approach has been recommended and used by many researchers for a further assessment. The actual amount of Pollutants known will vary depending on the application method, configuration, and equipment calibration and specific field conditions. Unrealistically high RQs are most prevalent in the preliminary assessment level of ERA. Unrealistic high ERA results may occur due to the inconsistent application of uncertainty factors and the continued overlooking of the biological definitions of RQs, as being more of a multiplicative of a benchmark by risk assessors when conducting screening-level risk assessments [56].

Besides, the lack of unification of deterministic approaches, including SQGs and RQ methods, may block the valid comparison between calculated risk values in the preliminary assessment of risk by different risk assessors. The risk level might differ significantly if separate database or risk assessment models are employed. To solve this problem, integrated approaches should be developed and used by researchers and public agencies to evaluate preliminary ecological risk.

ERA aims to support decision-making aiming to minimize the risk level in risk management effectively [51]. The inherent limitations of theses preliminary ERA approaches may exert a bad influence on the role of ERA results in risk management. Nevertheless, they are useful tools that can first guess the nature of a sediment pollution problem. Combined with engineering judgment, studies that employ the SQGs and RQ and appropriate field and laboratory sampling and testing, are essential tools in practice for sediment contamination, risk assessments and management.

## 4. Conclusions

This work evaluated the level of six PAEs, including DBP, BBP, DEHP, DnOP, DiNP and DIDP in the U-Tapao canal. Several deterministic approaches were used to analyze the potential ecological risk of individual PAEs and their mixture in the canal ecosystem. Deterministic risk assessment is a useful tool for the initial characterization of hazardous pollutants’ risks to ecosystems. The results revealed that SQGs and RQ, were not consistent with each other in the first tiered framework. The ecological risk assessment using SQGs suggested that DEHP and DBP levels in sediments posed a moderate risk on benthic organisms and the RQ method showed low or no threats on different trophic levels, including algae, crustacean and fish in the canal. However, amongst the target pollutants, DEHP toxicity in sediments seems to be the most critical issue. The combined risk caused by all the PAEs is significantly higher than any individual PAEs due to joint action. This baseline study would be helpful to the management and controlling of PAEs contamination in the aquatic environment. Moreover, to our knowledge, this is the first study to demonstrate a comparative screening ERA of individuals and a mixture of PAEs in an aquatic ecosystem. Furthermore, this work would serve as baseline data to support regulatory decision-making and for strategic pollutant mitigation measures in the canal ecosystem.

## Figures and Tables

**Figure 1 toxics-08-00093-f001:**
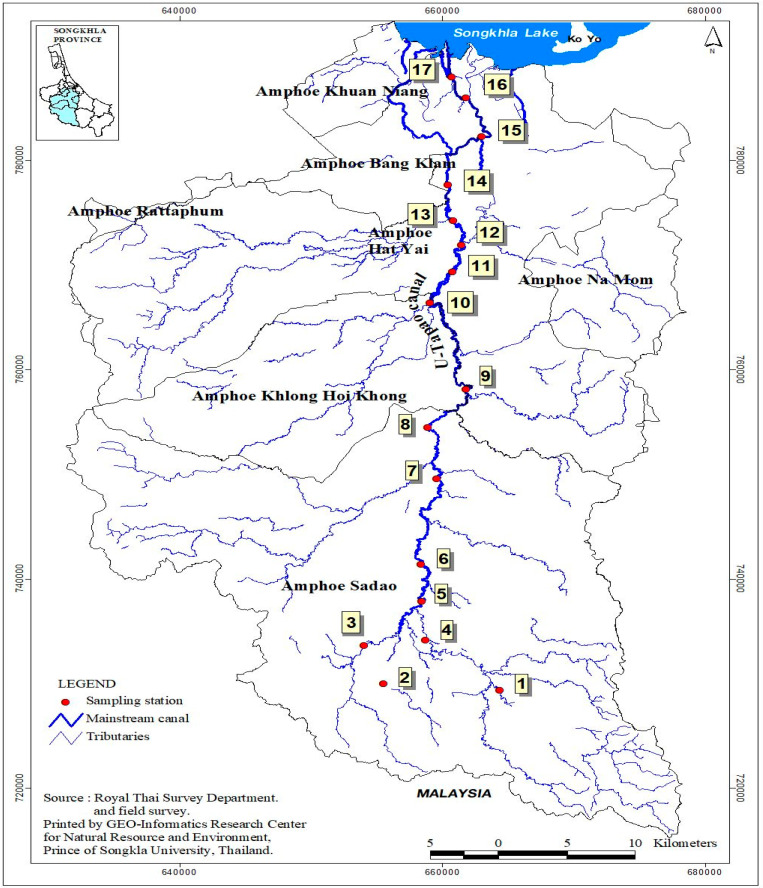
Map showing sampling sites of sediments in U-Tapao canal. (Source: Geoinformatics Research Center, Prince of Songkla University, 2019).

**Figure 2 toxics-08-00093-f002:**
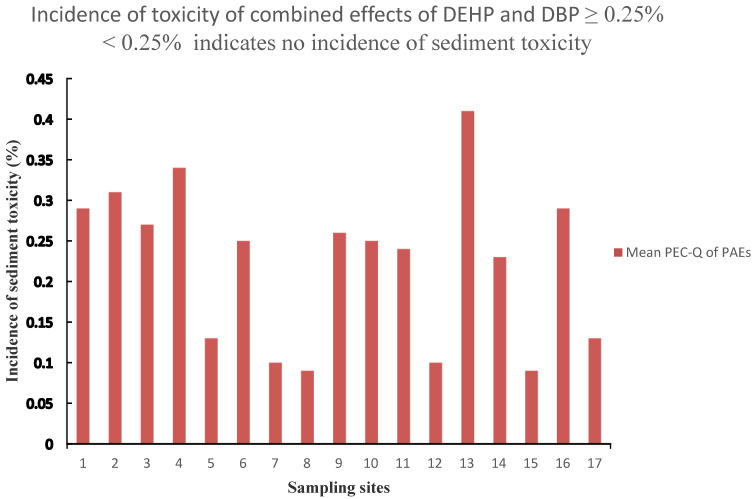
Distribution of the toxicity trend of DEHP and DBP mixture in U-Tapao canal.

**Table 1 toxics-08-00093-t001:** Quality Assurance /Quality Control parameters for the extraction and analysis of six targeted Phthalate esters (PAEs).

PAEs	Linearity R^2^	Target Ion (*m/z*)	Retention Time (min)	Recovery (%) (100 ng/g) *n* = 3	RSD (%)	LOQ *n* = 7 ng/mL	LOD *n* = 7 ng/mL
DBP	0.999	223, 205, 167	7.57	81	5.9	1.88	0.32
BBP	0.999	205, 149, 91	8.77	83	6.2	1.78	0.12
DEHP	0.999	279, 167, 149	9.29	93	7.2	2.98	0.45
DnOP	0.999	279, 261, 149	9.84	89	6.8	2.34	0.42
DiNP	0.999	293, 127	9.93	105	7.6	2.76	0.84
DIDP	0.999	307, 141	10.44	90	8.4	2.82	1.04

**Table 2 toxics-08-00093-t002:** Concentrations of PAEs and organic matter OM in sediments from U-Tapao canal (ng/g dw).

SITES	Latitude	Longitude	DBP	DEHP	DiNP	OM (%)
ST1	6.979739	100.463408	180	620	840	2.56
ST2	6.979740	100.463409	120	680	560	1.11
ST3	7.002145	100.455991	50	580	260	1.41
ST4	6.673192	100.433361	160	750	540	1.71
ST5	6.639564	100.436129	80	290	580	1.00
ST6	6.931202	100.439884	40	550	520	1.80
ST7	6.596520	100.486966	ND	220	380	1.20
ST8	7.108381	100.465011	ND	200	230	1.64
ST9	6.705086	100.433163	ND	560	ND	2.86
ST10	7.075167	100.475782	180	540	660	2.53
ST11	6.823206	100.437958	130	520	ND	4.00
ST12	6.779266	100.443868	90	220	ND	1.11
ST13	6.602124	100.406920	280	890	760	3.72
ST14	6.856377	100.464485	80	510	160	2.53
ST15	7.033356	100.452362	40	190	140	1.00
ST16	6.823206	100.437958	ND	640	ND	4.44
ST17	7.126859	100.455496	60	270	ND	2.01
Minimum			ND	190	ND	1.00
Maximum			280	890	840	4.44
Mean ± SD			88.82 ± 78	484 ± 223	333 ± 292	2.16 ± 1.09
Frequency of Detection			77%	100	71	-

ND: Nondetectable.

**Table 3 toxics-08-00093-t003:** Spearman correlation coefficient.

PAEs.	DBP	DEHP	DiNP	∑PAEs	OM (%)	pH
DBP	1					
DEHP	0.464	1				
DiNP	0.568 *	0.414	1			
∑PAEs	0.675 **	0.759 **	0.854 **	1		
OM (%)	0.171	0.484 *	−0.127	0.246	1	
pH	−0.129	−0.245	−0.201	−0.108	0.243	1

* Correlation is significant at the 0.05 level (2-tailed). ** Correlation is significant at the 0.01 level (2-tailed).

**Table 4 toxics-08-00093-t004:** Comparison of di-n-butyl phthalate (DBP), di-2-ethyl hexyl phthalate (DEHP) and di-isononyl phthalate (DiNP) levels in different countries (ng/g dw).

Location	DBP	DEHP	DiNP	Reference
Kaoshiung Harbor Taiwan	0.0–34.6	152.6–14,646.6	0.00–67,495.9	[35]
False Creek Vancouver, Canada	9320–63,900	7350–136,000	14,700–50,400	[41]
Jiulong River Estuary, China	1.6–92.8	4.3–394.7	ND-110	[39]
Dianbao River, Taiwan	400–1865	494–1947	361–1277	[40]
Jiulong River, China	3.0–230	7.00–1160	ND-470	[22]
ChangJiang River Estuary, China	340–7080	260–8550	NA	[42]
Chao Phraya River, Thailand	NA	<8340–14,500	NA	[16]
Eastern coast of Thailand	ND-800	ND-16500	NA	[17]
U-Tapao canal, Thailand	ND-280	190–890	ND-840	Present study

ND: non-detectable; NA: not available.

**Table 5 toxics-08-00093-t005:** Comparing the standard quality guidelines (SQGs) values of DEHP and DBP and levels in sediments of U-Tapao canal.

Standard Values of SQGs								Levels of DBP and DEHP in Sediments	
PAEs	TEL (ng/g)	PEL (ng/g)	NSQGQ	TEC (ng/g)	MEC (ng/g)	PEC (ng/g)	PEC-Q	DBP (ng/g dw)	DEHP (ng/g dw)
DBP	NA	NA	NA	2200	9600	17,000	≥0.25%	Level in sediment ND-230 TEC = 0% MEC = 0% PEC = 0% PEC-Q = 0.002–0.017%	190–890 TEL = 100% PEL = 0% NSQGQ = 0.51–1.16 TEC = 29% MEC = 6% PEC-Q = 0.17–0.81%
DEHP	182	2647	<0.2<2,>2	610	855	1100	≥0.25%		

NA: not available.

**Table 6 toxics-08-00093-t006:** Equilibrium partition (EqP) values of DEHP and DBP in U-Tapao canal.

PAEs	MPC (ng/g at 10% OM)	SRCeco (ng/g at 10% OM)	ERLs (ng/g at 10% OM)	DBP Level in Sediment (ng/g dw at 10% OM)	DEHP Levels in Sediment (ng/g dw at 10% OM)	∑PAEs Level in Sediments (ng/g dw at 10% OM)
DBP	2100	36,000	700	222–1081 MPC = 0% SRCeco = 0% ERLs = 41% sites		
DEHP	1000	10,000	1000		1219–6126 MPC = 100% sites SRCeco = 0% ERLs = 100%	
∑PAEs	1400	57,000	-			1464–12, 252 MPC = 100% of sites SRCeco = 0%

**Table 7 toxics-08-00093-t007:** Risk Quotient values for individual and mixture PAEs in sediments of U-Tapao canal.

PAEs/RQmix	Aquatic Biota	RQgeomean	RQmax
DBP	Algae	4.0 × 10^−2^	1.2 × 10^−2^
	Crustacean	3.0 × 10^−2^	1.0 × 10^−1^
	Fish	8.0 × 10^−2^	2.5 × 10^−1^
DEHP	Algae	1.5 × 10^−1^	2.7 × 10^−1^
	Crustacean	4.0 × 10^−2^	7.0 × 10^−2^
	Fish	2.0 × 10^−2^	3.0 × 10^−2^
DiNP	Crustacean	2.0 × 10^−2^	5.0 × 10^−3^
RQmixAlgae		1.9 × 10^−1^	3.9 × 10^−1^
RQmixCrustacean		7.0 × 10^−2^	1.8 × 10^−1^
RQ mix fish		1.0 × 10^−1^	2.8 × 10^−1^

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
