# Peer review of "Deterministic Assessment of the Risk of Phthalate Esters in Sediments of U-Tapao Canal, Southern Thailand"

_toxics, 2020, doi:10.3390/toxics8040093_

Round 1

Reviewer 1 Report

The manuscript entitled as "Deterministic ecological risk assessment of phthalate esters in sediments of U-Tapao Canal, Southern Thailand" presents the concentrations of phthalate esters in sediment in Southern Thailand and estimates the potential risks of them. Though this manuscript includes important results for environmental chemistry and risk assessments, the text is too verbose. And this manuscript contains many grammatical or basic errors. In addition, there is a lack of explanation on how to estimate sediment risk.

Here I list several examples of bad writing and errors in this manuscript. As the followings are not all flaws, I strongly recommend checking your writing throughout the entire manuscript.

  1. line 19. "results from the SQGs and RQ are not" -> "... were not"
  2. line 20. "caused moderate risk". The possible sediment risk was just estimated, not directly evaluated. So, "was estimated to cause" is better.
  3. line 23. The sentence "Moreover, the combined ..." is not necessary.
  4. Introduction is too verbose. Especially, paragraphs 5 and 6 can be shorten or written more simply.
  5. line 157. Write the material type of brown bottles (glass?). Contamination by laboratory tools is important for PAEs research.
  6. lines 190 to 198. Figure S1 in the supplementary file should be referred to. And Figure S1 should be replaced by a clearer one.
  7. line 243. NSQGQ is already spelled out (line 232).
  8. line 247. "PEC SQG concentration" -> "PEC is SQG concentration"?
  9. line 250. "equation three" -> "equation 3" 
  10. equation 3. What is PEC-Qs? Is this same as PEC-Q individual in the equation 2? If so, please use the same term.
  11. Materials and methods. I understand that the authors estimated the PNECsediment from PNECwater (Table S1) using EqP theory, but I cannot find how to calculate PNECsediment and which values were used for Koc in EqP approach in the Materials and Methods section.
  12. line 273. The sentence "Results of OM .." is not necessary in Materials and Methods section.
  13. Table 5. It's difficult to understand Table 5. 

Author Response

Hello Reviewer,

                    Thank you so much for your remarks and comment. I am very grateful because your review has improved our paper. Please see attachement.

Reviewer 2 Report

Authors present results of some PAEs in sediments of a freshwater system in Thailand collected in a sampling campaign with the aim of characterising the ecotoxicological risk for the aquatic communities. The risk, that a single compound of PAEs or their mixture posed to aquatic communities, is assessed with different methodological methods and results are compared.

Results are interesting and merit to be published, after being revised.

In particular, there is a lack of correspondence with the number quoted in the main text and the content of the quoted table. Please check throughout all the paper.

Authors reported that the concentration is normalised on dry weight (d.w.), please reported in the paper the concentration are in ng/g d.w. I think it is useful to avoid misunderstanding with potentially less expert readers (as students).

In chapter 2.4. Instrumental analysis by GC-MS

It is properly reported that SIM mode were selected as identification and quantification method. Please provide m/z of quantifier and qualifier ions.

No information are given on the use of internal standard and recovery standards. Please add information to improve the soundness of the results obtained.

Chapter 2.6. Ecological risk assessment of PAEs in sediments

Line 222 Pollutants in sediments can pose an adverse effect on benthic organisms via direct toxicity and  the alteration of the entire ecosystem[33].

I don’t agree with the second part of this sentence. It is not clear to me how pollutants in sediment can alter the entire ecosystems, particularly in this case, where concentration seems not to pose an unacceptable risk.

The consequences along the ecological hierarchic organisation are possible, but in this context the link with PAEs in sediment seems to me a Pindaric flight. Please revised this sentence.

Line 300. the spearman correlation matrix showed a low significant correlation

Please specify low. As statistical analysis are done, it is better to report the quantified value of this correlation and avoid the use of adjectives.

Line 444 In contrast, the maximum mixture concentration can produce 54% {Formatting Citation}.

This line needs to be rephrased.

Line 456-457. However, the sediment can be an extended/archived source of  anthropogenic compounds in the water column [49‐51]

In which way? Repartition of PAEs from sediment to water is expected to be low considering the high Log Kow.

Line 524 Fig. Three

Please change into number Fig. 3

Line 584-588 The RQ is intended to be used as rough indicators of potential risk and cannot be used to predict  how many aquatic organisms, including algae, crustacean, and fish that will die or experience adverse effects on their lifecycle or reproduction. RQ is not intended to predict the probability of a  fish or aquatic invertebrate receiving a lethal dose. RQ does not provide a definitive value for the  amount of pollutants that will be available to aquatic biotas such as algae, crustacean, and fish.

I do not agree with this paragraph. Any risk characterisation gives indication of the probability that an unacceptable effect can occur, not a quantification of how many individuals die. The ecotoxicological risk characterisation does not aim to protect any single individual, but the target is the maintenance of the structure and function of an ecosystems. Moreover the three standard organisms used In the assessment (algae, Daphnia and a fish) are in representation of all the communities of primary producer and consumer.

Please reconsider that sentence.

Supporting materials.

Please the figure 1 needs more editing effort

More information on the sampling campaign is needed, i.e. when it has been conducted.

Author Response

Hello Reviewer,

                     Thank you so much for your expert remarks and Comments. I am very grateful that your review has added value to the paper. Please see the attachement.

Round 2

Reviewer 1 Report

The quality of the manuscript has been improved. I think this paper can be published in Toxics.